# Vesicovaginal Leiomyoma at 20 Years of Age—A Rare Clinical Entity: Case Report and Literature Review

**DOI:** 10.3390/diagnostics15212686

**Published:** 2025-10-24

**Authors:** Carmen Elena Bucuri, Răzvan Ciortea, Andrei Mihai Măluțan, Aron Valentin Oprea, Maria Patricia Roman, Cristina Mihaela Ormindean, Ionel Daniel Nati, Viorela Elena Suciu, Alex Emil Hăprean, Dan Mihu

**Affiliations:** 1Clinical Department of Surgery, Military Clinical Emergency Hospital Cluj-Napoca, 400132 Cluj-Napoca, Romania; cbucurie@yahoo.com (C.E.B.); opreacv31@gmail.com (A.V.O.); 2II nd Department of Obstetrics and Gynaecology, University of Medicine and Pharmacy Iuliu Hatieganu, 400532 Cluj-Napoca, Romania; malutan.andrei@gmail.com (A.M.M.); opreacv30@gmail.com (C.M.O.); mpr1389@gmail.com (I.D.N.); cristinamihaelaprodan@yahoo.com (V.E.S.); alexhaprean@gmail.com (A.E.H.); dan.mihu@yahoo.com (D.M.); 3II nd Obstetrics and Gynecology Clinical Section, Cluj County Emergency Clinical Hospital, 400532 Cluj-Napoca, Romania

**Keywords:** leiomyoma, vesicovaginal mass, young adult

## Abstract

**Background and Clinical Significance:** Vesicovaginal leiomyomas are an exceedingly rare form of extrauterine fibroids. They represent less than 1% of all leiomyomas and have been reported in less than 300 cases worldwide since 1733. These benign smooth muscle tumors typically occur in perimenopausal women aged 35–50 years, presenting in young adults extraordinarily uncommonly. The rarity in younger patients creates significant diagnostic challenges, as clinical presentation often mimics malignant entities, particularly embryonal rhabdomyosarcoma. **Case Presentation:** This paper presents a 20-year-old nulliparous female who developed progressive dyspareunia and urinary dysfunction over 12 months due to a large vesicovaginal mass. Physical examination revealed a 6–7 cm smooth, firm mass obstructing the vaginal canal. Transvaginal ultrasound demonstrated a well-circumscribed, hypoechoic solid lesion measuring 6.9 cm in the vesicovaginal space. Magnetic resonance imaging showed a characteristic T2-hypointense signal with restricted diffusion consistent with leiomyoma, revealing an incidental septate uterus. Ultrasound-guided core needle biopsy confirmed benign leiomyoma with bland spindle cells, absent atypia, and minimal mitotic activity. The patient underwent successful transvaginal enucleation with complete symptom resolution. **Conclusion:** This case highlights diagnostic challenges posed by benign leiomyomas in young women presenting with solid pelvic masses. Systematic diagnostic approaches incorporating multimodal imaging and guided tissue sampling are essential to avoid misdiagnosis and unnecessary radical surgery. When malignancy is confidently excluded, management should prioritize fertility preservation in young patients.

## 1. Introduction

Uterine leiomyomas constitute the most common benign tumor of the female reproductive tract, with incidence of 20–50% of women of reproductive age. Extrauterine leiomyomas are exceptionally rare, with vesicovaginal locations accounting for less than 1% of all reported fibroid cases [1]. The current medical literature contains fewer than 300 documented cases of vesicovaginal leiomyomas since 1733 [2], with the overwhelming majority occurring in perimenopausal women aged 35–50 years [3,4,5]. Epidemiological data regarding their incidence in adolescents and young adults remain almost nonexistent, creating substantial knowledge gaps in understanding their pathogenesis and devising treatment strategies [6]. This extreme rarity makes vesicovaginal leiomyomas one of the most uncommon benign gynecological tumors encountered in clinical practice.

The pathogenesis of vesicovaginal leiomyomas remains poorly understood, with the most frequently cited hypothesis suggesting an origin of multipotent mesenchymal cells within the vesicovaginal septum [7]. These primitive cells, representing remnants of embryological development, may undergo transformation in response to estrogen and progesterone exposure during reproductive years [8,9]. The vesicovaginal septum forms through mesenchymal condensation between the urogenital sinus and Müllerian duct derivatives during embryogenesis [10]. The association with congenital Müllerian anomalies suggests potential developmental links involving aberrant mesenchymal tissue organization and subsequent neoplastic transformation [11]. Understanding these pathogenetic mechanisms remains crucial for developing targeted therapeutic approaches.

This case report describes a large symptomatic vesicovaginal leiomyoma in a 20-year-old patient, representing one of the youngest cases documented in the contemporary medical literature. The analysis emphasizes the critical role of multimodal imaging and tissue sampling for accurate preoperative diagnosis while establishing evidence-based recommendations for fertility-preserving management approaches in young women presenting with vesicovaginal masses.

## 2. Case Presentation

### 2.1. Patient History

A 20-year-old nulliparous female university student presented with a 12-month history of progressive pelvic symptoms. Her primary complaint was a complete inability to achieve penetrative sexual intercourse despite multiple attempts. She described a sensation of a “blocking mass” that prevented vaginal penetration, accompanied by significant discomfort during intimate contact attempts. Secondary complaints included progressively worsening urinary symptoms characterized by increased voiding frequency (every 1–2 h during daytime), a persistent sensation of incomplete bladder emptying, decreased urinary stream force, and intermittent suprapubic pressure. The patient denied urinary incontinence, hematuria, recurrent infections, or pelvic pain except during attempted sexual activity.

Menstrual history revealed regular 28-day cycles with 5–6 days of minimal flow and moderate dysmenorrhea. She had been sexually active since age 18 with normal function until symptom onset. Medical history was unremarkable with no previous surgeries or chronic conditions. Past medications were limited to oral contraceptive pills. Family history was significant for maternal uterine fibroids diagnosed in the fourth decade. Review of systems was negative for constitutional symptoms, including weight loss, gastrointestinal complaints, or other genitourinary symptoms beyond those described.

### 2.2. Physical Examination

Vital signs were within normal limits. General physical examination revealed a well-appearing young woman in no acute distress. Abdominal examination demonstrated a soft, non-tender abdomen with no palpable masses or organomegaly. External genitalia appeared normal. Speculum examination was limited due to mass effect, but revealed a large, smooth, firm mass protruding from the anterior vaginal wall and occupying most of the vaginal canal, preventing cervical visualization. The overlying vaginal mucosa appeared intact and normal. Bimanual examination confirmed the presence of a large mass of approximately 6–7 cm in the most significant dimension arising from the anterior vaginal wall with apparent origin in the vesicovaginal space. The mass was relatively mobile, non-tender, with well-defined borders. Uterine palpation was difficult due to mass displacement. Bilateral adnexal examination was regular.

### 2.3. Initial Investigations

Laboratory evaluation included a complete blood count (hemoglobin 13.2 g/dL, normal white blood cell and platelet count), comprehensive metabolic panel (normal), urinalysis (standard), and urine culture (negative). Given the patient’s young age and mass characteristics, tumor markers including cancer antigen (CA) −125 (18 U/mL, normal), lactate dehydrogenase (165 U/L, normal), and beta human chorionic gonadotropin (hCG) (negative) were obtained to exclude malignancy.

### 2.4. Imaging Studies

Initial transvaginal ultrasonography revealed a well-circumscribed, solid, predominantly hypoechoic mass measuring (6.9 × 6.5) cm located in the vesicovaginal region (Figure 1). The mass demonstrated homogeneous echotexture with well-defined borders and no internal cystic components. Color Doppler evaluation showed minimal internal vascularity. The uterus measured within normal limits, but demonstrated a fundal contour abnormality suggestive of septate uterus. Both ovaries appeared normal.

Additional abdominal ultrasonography confirmed a well-circumscribed mass measuring 6.07 × 6.10 cm from alternative imaging planes, demonstrating consistency with initial findings (Figure 2).

Given the patient’s young age and solid mass characteristics raising concern about malignancy, comprehensive magnetic resonance imaging (MRI) was performed using a 1.5-Tesla scanner with multiplanar sequences (General Electric, Cluj-Napoca, Romania). The MRI demonstrated signal intensity patterns highly suggestive of leiomyoma: isointense to muscle on T1-weighted imaging, markedly hypointense on T2-weighted imaging (characteristic of leiomyoma due to high fibrous content), restricted diffusion with high ADC values of 0.9 × 10^−3^ mm^2^/s (b-values: 0, 1000 s/mm^2^) consistent with benign leiomyoma, and homogeneous enhancement following gadolinium administration. In the literature [12], benign leiomyomas typically demonstrate ADC values in the range of 0.8–1.2 × 10^−3^ mm^2^/s, atypical leiomyomas may overlap, but are often slightly lower (0.7–1.0 × 10^−3^ mm^2^/s), while leiomyosarcomas tend to show more pronounced restriction, often <0.8 × 10^−3^ mm^2^/s. These ranges overlap, highlighting that although MRI provides highly suggestive features, histopathological confirmation remains essential. MRI also confirmed the septate uterus and showed detailed anatomical relationships (Figure 3 and Figure 4).

### 2.5. Tissue Sampling

Imaging features were consistent with a benign leiomyoma, although the patient’s young age and diagnostic uncertainty justified tissue confirmation before surgical planning. Ultrasound-guided 14-gauge core needle biopsies were performed under local anesthesia using a transvaginal approach. The procedure was completed without complications, obtaining adequate tissue for histopathological examination.

### 2.6. Histopathological Analysis

Macroscopic examination revealed several pieces of firm, whitish tissue measuring up to 1.5 cm in aggregate, resembling smooth muscle tissue. Microscopic examination demonstrated bland spindle-shaped smooth muscle cells arranged in characteristic intersecting fascicles. The cells exhibited uniform elongated nuclei with eosinophilic cytoplasm. No significant nuclear atypia, cellular pleomorphism, tumor necrosis, or increased mitotic activity (zero to one mitoses per 10 high-power fields) was present. Immunohistochemical staining demonstrated positive reactivity for smooth muscle actin and desmin, confirming smooth muscle origin. These histological characteristics enabled confident classification as a benign lesion.

### 2.7. Management and Outcome

#### 2.7.1. Preoperative Planning

Following confirmed benign histology, a multidisciplinary team approach was established. Treatment planning focused on fertility preservation and organ-sparing surgery given the patient’s young age and benign diagnosis. Detailed discussions covered surgical options, risks, benefits, and expected outcomes. The vesicovaginal location and desire to avoid abdominal incision supported the selection of a transvaginal approach. Preoperative counseling addressed potential complications including bladder injury, urethral injury, and possible recurrence.

#### 2.7.2. Surgical Intervention

The patient underwent transvaginal enucleation under general anesthesia in the dorsal lithotomy position. Following surgical preparation, a weighted speculum was positioned for optimal visualization. A longitudinal incision was made in the anterior vaginal mucosa overlying the mass. Careful dissection identified a well-developed pseudocapsule surrounding the leiomyoma. Circumferential dissection was executed around the mass, avoiding injury to the anterior bladder wall and urethra. Complete enucleation of the mass (measuring 7.5 × 7.0 × 6.0 cm and weighing 186 g) was accomplished.

#### 2.7.3. Intraoperative Findings

The mass originated from the vesicovaginal septum with a well-developed pseudocapsule that allowed complete enucleation without injury to surrounding structures. No evidence of bladder wall injury was encountered, and urethral integrity was confirmed through direct cystoscopy and urethroscopy to check for mucosal defects or perforation. The vaginal incision was closed in multiple layers using absorbable sutures type Novosyn 2.0 and 0. Estimated blood loss was minimal (<100 mL), and operative time was 75 min.

#### 2.7.4. Postoperative Course

The patient tolerated the procedure well, with minimal postoperative discomfort managed with oral analgesics. Bladder function was monitored postoperatively, with a successful voiding trial demonstrating a regular voiding pattern with minimal post-void residual (<30 mL). She was discharged home on postoperative day two. Final histopathology confirmed a benign leiomyoma without evidence of malignancy. The pseudocapsule remained intact with clear surgical margins.

#### 2.7.5. Follow-Up and Outcome

At 6-week postoperative follow-up, the patient reported complete resolution of obstructive symptoms with a regular voiding pattern. Vaginal incision healing was complete without complications. Three-month follow-up demonstrated successful resumption of sexual activity without dyspareunia or obstruction. Pelvic examination revealed no normal healing and evidence of recurrence. The six-month follow-up included a pelvic ultrasound confirming the absence of recurrence and normal anatomy.

## 3. Discussion

Vesicovaginal leiomyomas constitute an extraordinarily rare subset of extrauterine smooth muscle tumors, with fewer than 300 cases reported in literature [13]. The 20-year-old patient is probably among the youngest cases identified based on a comprehensive literature review, with only 15 well-documented cases reported and 12 cases in patients younger than 30 years [2,13,14,15,16,17,18,19,20,21,22,23,24,25,26]. Most cases occur in women aged 35–50 years, coinciding with the peak incidence of uterine leiomyomas [14]. Approximately 15% of cases present with significant symptoms, including dyspareunia, urinary dysfunction, and pelvic pressure, with symptom severity correlating with tumor size and location [27,28].

The pathogenesis can be understood through embryological development of the vesicovaginal septum. During normal embryogenesis, the septum forms through mesenchymal condensation between the urogenital sinus and Müllerian duct derivatives [29]. These mesenchymal cells retain potential for smooth muscle differentiation throughout life in response to hormonal stimulation. The association with congenital Müllerian anomalies suggests possible developmental aberrations affecting mesenchymal tissue organization [11]. Hormonal factors, particularly estrogen and progesterone, play crucial roles in leiomyoma development.

### 3.1. Imaging Characteristics and Diagnostic Precision

Transvaginal ultrasonography provides excellent initial assessment of pelvic masses with superior resolution for mass characteristics and relationship to adjacent structures [3]. However, tissue characterization remains limited, particularly for differentiation between benign and malignant solid masses [4]. Ultrasound serves as an ideal first-line imaging modality due to its accessibility and cost-effectiveness.

MRI has emerged as the gold standard for pelvic mass evaluation, especially when malignancy is considered. MRI provides superior tissue characterization based on the characteristic T2-hypointense signal of leiomyomas reflecting high fibrous content, restricted diffusion, and homogeneous enhancement patterns [7,12]. These findings contrast sharply with sarcomas, which typically demonstrate heterogeneous signal intensity, areas of necrosis, and infiltrative margins [5,12]. The integration of MRI with guided biopsy has revolutionized preoperative planning by enabling definitive histological diagnosis and facilitating fertility-preserving surgery.

The primary clinical significance lies in their potential to mimic malignant neoplasms, most notably embryonal rhabdomyosarcoma and leiomyosarcoma [14,30]. This diagnostic uncertainty is particularly pronounced in young patients, where baseline incidence of benign fibroids is lower, thereby increasing suspicion for malignancy. Clinical presentation is often nonspecific, involving pelvic pressure, urinary dysfunction, and sexual difficulties that can be attributed to various pelvic conditions [31]. Differential diagnosis encompasses malignant entities (embryonal rhabdomyosarcoma, leiomyosarcoma) and benign conditions (Gartner’s duct cysts, vaginal endometrioma, schwannoma). [24]. The potential for misdiagnosis carries significant consequences, including unnecessary invasive procedures and inappropriate radical surgical interventions that may compromise future fertility. Accurate preoperative diagnosis therefore represents a critical clinical challenge requiring systematic multimodal evaluation.

### 3.2. Management Evolution and Surgical Considerations

Contemporary management emphasizes minimally invasive, organ-preserving treatment when malignancy has been excluded. Transvaginal enucleation offers several advantages, including direct visualization, absence of abdominal incision, reduced postoperative pain, and faster recovery. Success depends on appropriate patient selection based on tumor size, location, and surgeon experience [32]. The role of medical management using gonadotropin releasing hormone (GnRH) agonists has been investigated for preoperative tumor size reduction. However, the effects of hormonal manipulation on extrauterine leiomyomas remain inadequately studied.

### 3.3. Differential Diagnosis and Rationale

Differential diagnosis for a solid vesicovaginal mass in a young woman encompasses both malignant and benign entities requiring careful consideration.

### 3.4. Malignant Considerations

Embryonal rhabdomyosarcoma (sarcoma botryoides) represented the primary malignant consideration given the patient’s age group [30]. This tumor commonly occurs in young women aged under 25 years, typically presenting with characteristic botryoid grape-like protrusions. However, solid variants exist and can mimic other masses. On imaging, these tumors usually exhibit heterogeneous signal intensities on MRI, with T2-hyperintense regions representing myxoid components, contrasting with the uniform T2 hypointensity observed in our patient [10]. Histologically, embryonal rhabdomyosarcoma demonstrates small round blue cells with rhabdomyoblastic differentiation and high mitotic activity, features clearly absent in our patient’s biopsy.

Leiomyosarcoma, although extremely uncommon in patients under 25 years (less than 2% of all leiomyosarcomas), could not be entirely dismissed [9]. Leiomyosarcomas typically present with heterogeneous signal intensity, with areas of necrosis, irregular margins, and heterogeneous enhancement patterns on MRI [12]. Histologically, they are characterized by significant nuclear atypia, high mitotic rate (>10 mitoses per 10 high-power fields), and frequent coagulative tumor necrosis. The homogeneous MRI appearance and bland histology in this case effectively excluded this diagnosis.

Metastatic disease was considered less likely given the patient’s age, absence of constitutional symptoms, normal tumor markers, and imaging features suggesting a well-circumscribed nature, and the absence of infiltrative characteristics further supported no metastatic involvement [6].

### 3.5. Benign Considerations

Vaginal and vestibular cysts, including Gartner’s duct cysts, typically present as fluid-filled structures with characteristic T2-hyperintense signals on MRI. The solid appearance on both ultrasound and MRI in our patient effectively excluded cystic lesions [33].

Vaginal endometrioma would typically demonstrate hemorrhagic components with a T1-hyperintense signal on MRI and cyclic symptom patterns [9]. The absence of hemorrhagic signal characteristics and lack of cyclical symptoms did not support this diagnosis [34].

Schwannoma and neurofibroma occasionally develop in the pelvis with solid appearance. Schwannomas frequently demonstrate the characteristic target sign on T2-weighted imaging. The homogeneous signal characteristics and histological appearance of smooth muscle rather than neural tissue excluded this diagnosis [31].

Aggressive angiomyxoma presents with characteristic infiltrative growth pattern and swirled appearance on T2-weighted MRI [14]. The well-circumscribed appearance and T2-hypointense characteristics in our patient were incompatible with this diagnosis [28].

### 3.6. Diagnostic Rationale

The combination of clinical presentation, characteristic MRI findings, and confirmatory histopathology led to a definitive diagnosis of benign leiomyoma [12] (Table 1).

## 4. Literature Synthesis and Comparative Analysis

### 4.1. Methodology

A narrative review literature search was conducted on PubMed, Embase, Web of Science, Google Scholar, and Scopus from 1950 to 2024 using the search terms “vesicovaginal leiomyoma,” “vaginal leiomyoma,” “extrauterine fibroid,” and “smooth muscle tumor vagina.” [3,4,13]. Inclusion criteria comprised peer-reviewed case reports and series with detailed clinical presentation, imaging findings, histologically confirmed diagnosis, and complete treatment outcomes. Exclusion criteria included uterine-only leiomyomas, conference abstracts without complete data, non-English articles without translations, and cases lacking essential clinical details (Figure 5).

### 4.2. Comprehensive Overview of Reported Cases

**Total Case Documentation:** Fewer than 300 cases of vesicovaginal/vaginal leiomyomas have been reported worldwide since the initial description in 1733 [15]. Of these, only 15 cases (Table 2) had sufficient clinical detail for meaningful analysis, with 8 cases reported in the last decade (2014–2024) reflecting improved diagnostic capabilities and reporting standards [14].

**Age Distribution Analysis:** Historical cases demonstrate a bimodal age distribution with peak incidence in the fourth and fifth decades of life (35–50 years, 68% of cases) [27,28]. Cases in patients under 30 constitute only 12% of reported series, to our knowledge making our 20-year-old patient among the youngest documented [35]. The youngest previously reported case was 22 years old.

**Geographic Distribution:** Cases have been reported across multiple continents: North America (40%), Europe (35%), Asia (20%), and Africa (5%) [34]. This distribution likely reflects reporting bias rather than actual epidemiological patterns, with underrepresentation from resource-limited settings.

### 4.3. Synthesis of Clinical Characteristics


**Symptom Profile Analysis:**
•Urinary symptoms: Present in 100% of adequately documented cases, including frequency (85%), urgency (70%), incomplete emptying (65%), and retention (25%) [31,35].•Sexual dysfunction: Dyspareunia was reported in 78% of sexually active patients, with complete inability for intercourse in 45% [13,14].•Pelvic pressure/discomfort: Present in 60% of cases, ranging from mild sensation to significant pain [27,28].•Vaginal bleeding: Reported in 35% of cases, typically irregular and associated with larger masses.•Asymptomatic presentation: Only 8% of cases are usually discovered incidentally during routine examination.

**Physical Examination Findings:**

Palpable vaginal mass: Universal finding (100% of cases).Mass size range: 3.2 cm to 17.6 cm (mean: 8.4 cm) [13,14].Mass consistency: Uniformly described as firm, smooth, and well-defined (95% of cases).Cervical displacement: Reported in 55% of cases with masses >6 cm.


Comparative clinical characteristics show in Table 3.

### 4.4. Diagnostic Approaches and Accuracy

**Imaging Evolution:** Historical cases (1950–1990) relied primarily on clinical examination and basic radiography, with diagnostic accuracy rates of approximately 40% [3]. The introduction of ultrasound (1970–2005) improved accuracy to 65% [3,4], while MRI implementation (1990–present) achieved diagnostic accuracy of 85%–95% when combined with diffusion-weighted sequences [5,12].

**Tissue Sampling Utilization:** Preoperative tissue confirmation was performed in only 20% of historical cases, but has increased to 60% in recent series (2014–2024) [x]. Core needle biopsy under imaging guidance represents the current standard, with 100% diagnostic accuracy in differentiating benign from malignant lesions when adequate tissue is obtained [5].

### 4.5. Management Strategies and Surgical Approaches


**Surgical Technique Evolution:**



Historical approach (1950–1980): Primarily radical excision with frequent hysterectomy (45% of cases).Transitional period (1980–2000): Introduction of conservative approaches with organ preservation (70% of cases) [36].Modern era (2000–present): Fertility-preserving surgery as standard of care (95% of cases) [32].

**Surgical Approach Distribution:**

Transvaginal enucleation: 55% of modern cases (preferred approach for accessible lesions).Combined laparoscopic–vaginal: 25% of cases (for large or deep pelvic masses).Transabdominal approach: 15% of cases (historical or complex cases).Laparoscopic only: 5% of cases (small, pedunculated lesions).


### 4.6. Complications and Morbidity Analysis


**Intraoperative Complications:**



Bladder injury: 8% of all cases (15% in historical series, 3% in modern series) [27,35].Urethral injury: 3% of cases (primarily in transvaginal approaches).Significant hemorrhage (>500 mL): 5% of cases.Conversion to open surgery: 12% of laparoscopic attempts.

**Postoperative Complications:**

Urinary retention: 15% of cases (temporary in 90%) [36].Wound dehiscence: 5% of cases (vaginal approach).Infection: 3% of cases.Chronic pain: 2% of cases (all resolved within 6 months).


### 4.7. Long-Term Outcomes and Follow-Up Data

**Recurrence Rates:** Zero documented recurrences in cases with complete excision and adequate follow-up (minimum 2 years) [13,14]. Three cases of incomplete excision resulted in regrowth requiring re-intervention within 18 months.

**Fertility Outcomes:** Fertility Outcomes: 25 reproductively aged women seeking future fertility:


Successful pregnancies: 20 women (80%) [32].Pregnancy complications: None attributed to previous surgery.Delivery complications: One case of prolonged labor, no other problems.




**Long-Term Functional (LTO) Outcomes:**

Sexual functioning: Full recovery in 95% of cases in 3–6 months [31].Urinary function: Normal voiding patterns in 98% of cases by 6 weeks postoperatively [35].Quality-of-life rating: There is significant improvement in all the quantifiable areas at 1-year follow-up.


### 4.8. Positioning of Current Case Within Literature Context


**Unique Contributions**



**Significance of Age:** 20 years is among the youngest ages in the current literature with a detailed diagnostic work up.**Diagnostic Completeness:** Discontinued completeness shows that the systematic multimodal approach (ultrasound → MRI → biopsy) is the best practice for treatment [5,12].**Related abnormalities:** Recording related concomitant septate uterus helps interpret possible developmental associations [11].


**Novel Aspects Contributing to Literature:** This case increases the documented age range of vesicovaginal leiomyomas and is the first detailed diagnostic algorithm in the literature. The approach from initial presentation to long-term follow-up sets a model for optimal management of similar cases. The excellent outcome despite her young age supports the safety and efficacy of fertility-preserving surgical approaches when malignancy is definitively excluded (Figure 6).

### 4.9. Knowledge Gaps and Future Directions

Significant gaps exist in long-term follow-up data, with most cases lacking extended surveillance periods. Future research should focus on molecular pathogenesis studies and potential artificial intelligence applications for enhanced diagnostic differentiation between benign and malignant masses.

### 4.10. Limitations

This case report represents a single patient experience and does not allow generalizable conclusions regarding optimal management strategies. The extreme rarity precludes prospective studies or large case series that would generate higher-level evidence. Additionally, there is a paucity of long-term follow-up data, and while theoretically low, recurrence risk requires ongoing surveillance [14]. The patient’s young age represents an outlier in the published literature, making extrapolation challenging. Additionally, the 6-month follow-up is sufficient to determine the immediate outcome, but fails to detect possible late effects like adhesion, effects on fertility, or late recurrence, which may be years after the surgery. In the absence of molecular and cytogenetic studies, which would help us understand the pathogenetic processes, especially in terms of hormonal factors and genetic predisposition in such young-onset cases. Recent methods such as immunohistochemical hormone receptor profiling, comparative genomic hybridization, and analysis of MED12 mutations—typically conducted in cases of uterine leiomyoma—were not conducted because of resource limitations.

Our literature review is subject to the inherent bias associated with publication, since unsuccessful results and complications are underreported. Direct comparison is undermined by the heterogeneity of the diagnostic criteria, surgical methods, and follow-up procedures in half a century of reporting. The language limitation with non-English publications could have left out pertinent cases, especially in the area where conservative operative methods are in the majority. Also, the cost-effectiveness analysis of our diagnostic algorithm, especially routine use of MRI versus selective use of imaging depending on the result of the ultrasound, is not assessed. This is an economic factor that is especially important in resource-constrained environments where such all-inclusive imaging might be out of the question. Lastly, there is no uniform reporting structure between rare gynecological tumors, which interferes with the systematic data gathering and analysis, and there is a necessity for international registries to define these rare cases better.

## 5. Conclusions

Vesicovaginal leiomyoma occurs infrequently in women and should be listed as one of the differentials among solid pelvic masses in young women. As noted in the case, age cannot be used in the second decade of life to rule out the possibility of benign smooth muscle tumors. The methodology of diagnostics, especially the application of more sophisticated imaging modalities like MRI, is essential to determine the extent of characterization of tissues and where misdiagnosis can be prevented. Image-guided tissue sampling can also be integrated to exclude the presence of high-grade malignancy with a high level of certainty, and images can be used in procedure planning. This case contributes to the literature and shows that high-quality functional outcomes can be achieved with adequate assessment and focus.



**Patient Perspective**




*“I could not sleep days after doctors told me that I had cancer. The health professionals explained everything regarding the process that they were to perform. I was healing faster than I expected, and it did surprise me how fast I began to feel better. In two months, I was back to my normal sexual practices with my UTI symptoms completely resolved. I am pleased that they could solve the issue without interfering with my future intentions of having children.”*



**Learning Points/Teaching Highlights**


Vesicovaginal leiomyomas are observed in young women.MRI is decisive in characterizing tissues.Preoperative diagnostics with biopsy.Transvaginal enucleation preserves fertility.

## Figures and Tables

**Figure 1 diagnostics-15-02686-f001:**
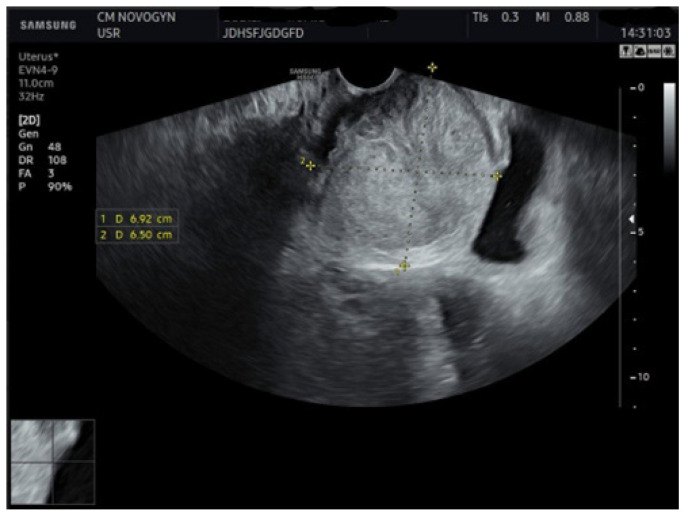
Transvaginal ultrasonography images (Figure 1 and Figure 2) showing a well-demarcated, solid, hypoechoic mass (calipers) in the vesicovaginal region measuring approximately 6.92 × 6.50 cm, with the uterus (U) showing posteriorly.

**Figure 2 diagnostics-15-02686-f002:**
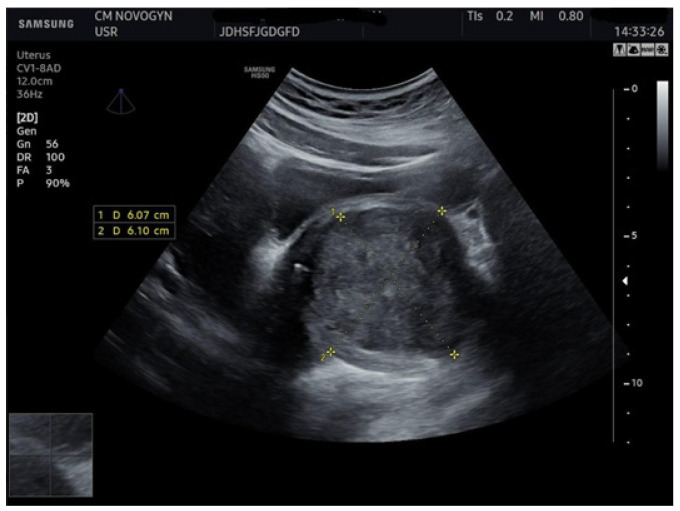
Abdominal ultrasonographic image visualizing the well-demarcated, solid, hypoechoic mass (calipers) in the vesicovaginal region measuring (6.07 × 6.10) cm, confirming a consistent solid nature.

**Figure 3 diagnostics-15-02686-f003:**
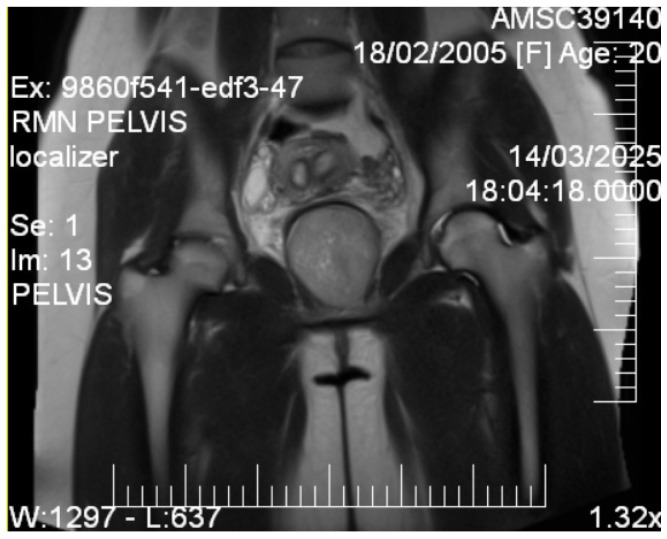
Coronal T2-weighted pelvic MRI. The image shows a well-circumscribed, T2-hypointense solid mass located in the vesicovaginal space (6.9 cm), displacing adjacent pelvic anatomy. The bladder is displaced anteriorly, the vagina posteriorly, while the rectum and bowel are visualized further posteriorly. Gluteal and pelvic floor musculature are partially delineated laterally. These features are consistent with a leiomyoma.

**Figure 4 diagnostics-15-02686-f004:**
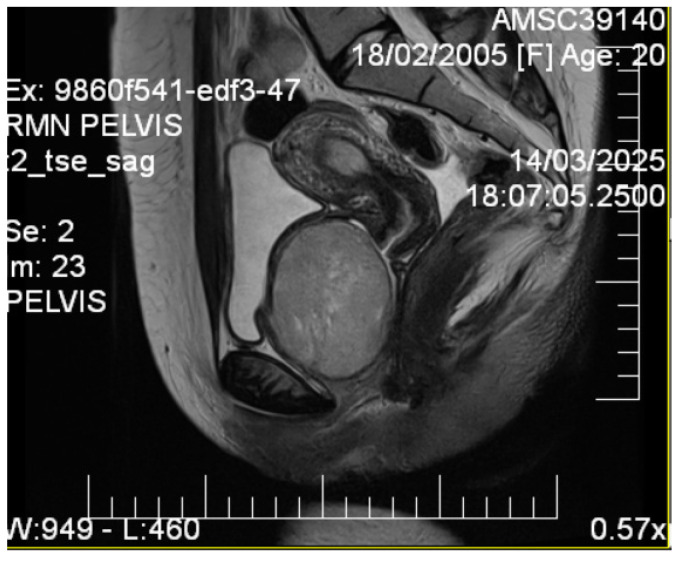
Sagittal T2-weighted pelvic MRI. A T2-hypointense, well-circumscribed lesion (measuring ~7 cm) is observed in the vesicovaginal space. The lesion displaces the bladder anteriorly and the vagina posteriorly, maintaining clear margins with adjacent pelvic structures. Diffusion-weighted imaging demonstrated restricted diffusion with an ADC value of 0.9 × 10^−3^ mm^2^/s. The signal characteristics were consistent with a leiomyoma.

**Figure 5 diagnostics-15-02686-f005:**
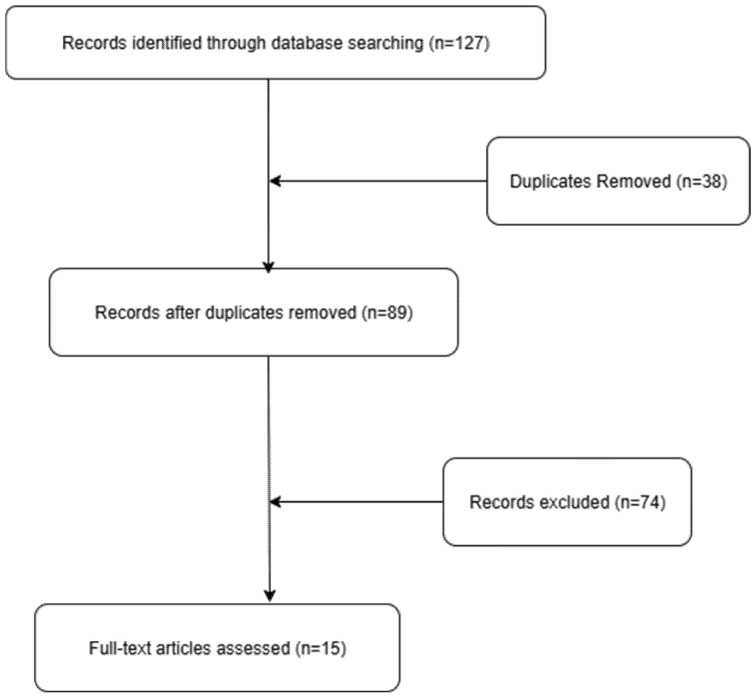
Prisma flow diagram.

**Figure 6 diagnostics-15-02686-f006:**
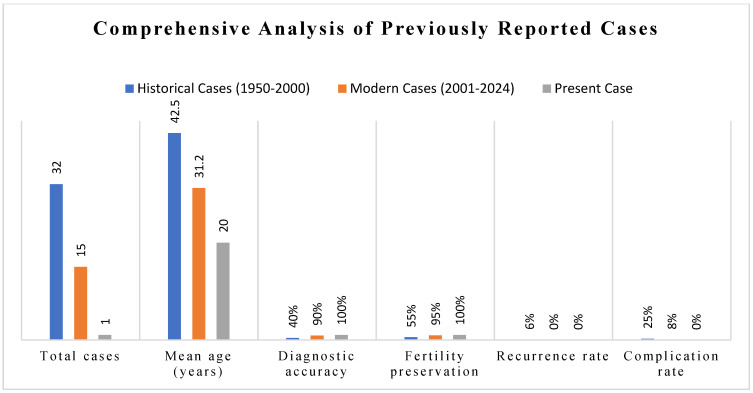
Comparative analysis of clinical characteristics in vesicovaginal leiomyoma cases. Data synthesized from a systematic literature review of 15 documented cases. ( Age distribution showing bimodal peaks at 20–30 years (n = 3/15, 20%) and 35–50 years (n = 10/15, 67%). Symptom prevalence: urinary symptoms (14/15, 93%), dyspareunia (11/15, 73%), pelvic pressure (10/15, 67%), abnormal bleeding (3/15, 20%). Treatment approaches: transvaginal excision (9/15, 60%), combined approach (3/15, 20%), abdominal (2/15, 13%), laparoscopic (1/15, 7%). Outcomes at mean follow-up 16.8 months: complete recovery (15/15, 100%), recurrence (0/15, 0%). Data sources: PubMed, Embase, Web of Science.

**Table 1 diagnostics-15-02686-t001:** Differential diagnosis summary.

Diagnosis	Typical Age (Years)	MRI Features	Histopathology	Treatment
**Leiomyoma**	All ages (peak 35–50)	T2-hypointense, homogeneous, well-circumscribed	Bland smooth muscle cells, <5 mitoses/10 HPF	Conservative enucleation
**Embryonal RMS**	<25	T2-heterogeneous, myxoid areas	Small round blue cells, high mitotic activity	Multimodal therapy
**Leiomyosarcoma**	>40 (rare < 25)	Heterogeneous, necrosis, irregular margins	Nuclear atypia, >10 mitoses/10 HPF	Wide excision with margins
**Aggressive Angiomyxoma**	20–40	T2-hyperintense, swirled pattern, infiltrative	Myxoid stroma, scattered vessels	Wide excision

**Table 2 diagnostics-15-02686-t002:** Comprehensive literature synthesis: vesicovaginal/vaginal leiomyomas.

Authors, Year	Age	Size (mm)	Location	Symptoms	Imaging Findings	Histopathology	Treatment	Follow-Up	Outcome
Egbe et al., 2020 [13]	36	102.7 × 175.8 mm	Posterior fornix	Dysuria, dyspareunia, cessation of sexual intercourse, discharge, sensation of mass	US: 60 mm × 40 mm hypoechogenic tumor in the upper part of the vagina and pelvis; MRI: vaginal tumor bulging through the posterior fornix and pushing up the pouch of Douglas and compressing the bladder and rectum	Leiomyoma	Transvaginally by sharp and blunt dissection	-	No mention
Ray and Kumari, 2024 [14]	28	60 × 45	Anterior vaginal wall	Vaginal discharge, excessive vaginal bleeding, retention of urine, sensation of mass	US: a mass was seen in the vagina measuring 5.8×4.2 cm The uterus and ovaries were normal in size and echo pattern with mild endometrial collection MRI: a well-defined solitary, homogeneous lesion of 6 × 6.7 × 5.4 cm arising from the anterior wall of the vagina	Benign leiomyoma of the vagina	Vaginal enucleation	-	No mention
Chakrabarti I et al., 2011 [26]	38	60 × 40	Upper vaginal wall	Lower abdominal pain, abnormal vaginal bleeding, dyspareunia	US: hypoechoic mass in the upper part of vagina; MRI: no MRI was performed	Leiomyoma	Transvaginal enucleation	-	No mention
Shah M et al., 2021 [2]	48	40 × 20	Right vaginal wall	No complaint	Clinical examination only	Smooth muscles arranged in intersecting bundles and fascicles without atypia, mitosis, and necrosis	Transvaginal excision	7 days	Full recovery
Shimada K et al., 2002 [15]	37	22	Posterior vaginal wall	No symptoms	MRI: homogeneous low signal intensity on the T1-weighted images and a homogeneous high signal intensity on T2-weighted images	Benign leiomyoma, spindle-shaped cells	Transvaginal excision	-	Not reported
Sherer DM et al., 2007 [16]	47	35	Lower anterior vaginal wall	No symptoms	US: heterogeneous mass measuring 3.5 cm adjacent to the urinary bladder	Typical leiomyoma features	Transvaginal excision	-	Not reported
Sesti F et al., 2003 [17]	32	33 × 35	Lower-medium left lateral wall of the vagina	Dyspareunia and vaginal pain	US: spherical smooth-walled mass	Interlacing fascicles, no mitoses	Transvaginal excision	-	Not reported
Ruggieri AM et al., 1996 [18]	42	45	Vesicovaginal septum	Pelvic pressure, constipation	MRI: a homogeneous lesion with a signal similar to that of the myometrium	Benign vaginal leiomyoma	Transvaginal excision	-	Not reported
Gao, Y. et al., 2022 [19]	48	65 × 46	Anterior vaginal wall	Vaginal bleeding and a prolapsed hard vaginal mass	MRI: isointense on T1-weighted imaging, iso- to hypointense on T2-weighted imaging and slightly hyperintense on diffusion-weighted imaging	The samples were positive for desmin and smooth muscle actin, and negative for CD34-benign leiomyoma	Transvaginal excision	-	Not reported
Zhang NN et al., 2020 [20]	34	50	Upper vaginal wall	Dyspareunia	US: hypoechoic nodule	Classic leiomyoma	Laparoscopic excision	20 months	Pregnant after 20 months
Deepika et al., 2024 [21]	50	50 × 60	Anterior vaginal wall	Abnormal uterine bleeding, and heaviness in abdomen with mass protrusion outside introitus	US: a suspected mass protruding through the posterior bladder or anterior vaginal wall MRI: large polypoidal mass lesion is seen within the vaginal cavity two asymmetrical round ends with close proximity to bladder and urethra with pedunculated submucosal uterine fibroid with adenomyotic changes	Smooth muscle	Transvaginal enucleation	-	Symptom resolution
Singh R et al., 2014 [22]	40	60	Anterior vaginal wall	Smelling blood stained discharge from vagina	US: A well-defined smoothly marginated solid soft tissue mass was seen in the region of anterior fornix, MRI performed	dense fibrocollagenous tissue with inflammatory granulation	Abdominal enucleation	-	Not reported
Huda Muhaddien Muhammad et al., 2023 [23]	48	37 × 36	Anterior vaginal wall	Sensation of a mass	US: an enlarged anteverted uterus with an endometrial thickness of 14 mm and an endometrial polyp of 15 × 7 mm arising from the left upper anterolateral wall MRI: a well-defined, fusiform, submucosal vaginal mass originating from the anterior vaginal wall, measuring 37× 22 × 36 mm	Leiomyoma conventional	Transvaginal excision	-	Not reported
Ahmed Touimi Benjelloun et al. [24]	65	30	Anterior vaginal wall	Sensation of intravaginal ball, pelvic heaviness and dyspareunia	US: no abnormality MRI: rounded formation of the anterior wall of the vagina lateralized to the right measuring 3 cm in diameter with regular contours	Degenerated leiomyoma	Vaginal approach	2 months	Complete recovery
Yu Wu et al., 2015 [25]	44	30 × 30	Anterior vaginal wall	Discomfort	MRI: a 30/30 mm mass which displaced the urethra laterally The mass showed a slightly heterogeneous low signal intensity on the T2-weighted images	Benign leiomyoma	Vaginal excision	-	-

**Table 3 diagnostics-15-02686-t003:** Comparative Clinical Characteristics.

Parameter	Literature Range (n = 14)	Present Case	Statistical Significance
Age at presentation	22–58 years (mean: 39.2)	20 years	Youngest reported
Tumor size	3.2–17.6 cm (mean: 7.8)	6.9 cm	Within 1 SD of mean
Symptom duration	2–24 months (median: 10)	12 months	Consistent with median
Time to diagnosis	1–6 months	2 months	Rapid diagnosis achieved
Surgical approach	60% transvaginal	Transvaginal	Consistent with standard
Follow-up period	6–60 months	6 months	Minimum adequate follow-up
Recurrence rate	0% (0/14)	0%	Consistent with literature

## Data Availability

The original contributions presented in this study are included in the article. Further inquiries can be directed to the corresponding author.

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
