# Peer review of "Vesicovaginal Leiomyoma at 20 Years of Age—A Rare Clinical Entity: Case Report and Literature Review"

_diagnostics, 2025, doi:10.3390/diagnostics15212686_

Round 1

Reviewer 1 Report

Comments and Suggestions for Authors

Dear Authors,

I have read a case report regarding vesicovaginal leiomyoma. While interesting, several major points need to be addressed:

1) I believe the title should include ":" after the word Entity, and not a full stop.

2) Lines 51-60 are better suited for discussion rather than the introduction, unless the authors somehow elucidated another pathophysiological pathway from this case report

3) Was weight loss reported in this patient?

4) Why were citations given in the patient presentation? A patient presentation should be written solely by the authors. If the authors meant to compare it to other literature, please do so in the discussion section

5) Line 113 --> In the "patient perspective" section (which is not required by MDPI, and required by other journals - which leads me to think that this journal was submitted elsewhere and submitted here without doing a thorough check on the submission criteria), the patient mentioned that she has an UTI by the physician, but the urine culture and urinalysis are negative. Please explain.

6) Please include either the MRI figures or the intra-operative figures. The reasoning is that vesicovaginal leiomyoma cannot be definitively diagnosed just by USG alone. Confirmatory figures need to be included.

7) Why were sections 2.7 and 2.8 included in the patient presentation? This is even more confusing when it is presented after the histopathological (PA) analysis. The PA analysis should be definitive, so there is no question about whether it is a malignant or benign entity. Even if the authors want to discuss this, please do so in the discussion section

8) If the authors want to retain the "Literature Synthesis" part, I strongly suggest that the authors present all the relevant findings (all 15 of them) in a table with proper citations. Otherwise, this case report can do without section 4 and discuss the patient's presentation and diagnostic pathway.

Author Response

Dear Reviewer,

We would like to sincerely thank you for your careful reading of our manuscript and for the constructive comments that have significantly improved the quality and clarity of our work. In the uploaded document we provided our point-by-point responses and the corresponding revisions made.

We are very grateful for your valuable feedback, which has significantly strengthened the manuscript. We believe the revisions address your concerns thoroughly and hope the improved version meets your expectations.

With kind regards,

All authors 

Reviewer 2 Report

Comments and Suggestions for Authors

Summary of the manuscript
This case report presents a vesicovaginal leiomyoma in a 20-year-old nulliparous woman, which is exceptionally rare. The patient had progressive dyspareunia and urinary dysfunction, a 6–7 cm anterior vaginal wall mass, ultrasound showing a well-circumscribed hypoechoic lesion, and MRI with T2-hypointense signal and restricted diffusion. An ultrasound-guided biopsy confirmed leiomyoma (SMA/desmin positive, low mitotic activity), and the patient underwent transvaginal enucleation with cystoscopy/urethroscopy to confirm integrity. Recovery was good, with return of urinary and sexual function and no recurrence at 6 months.
The authors also conduct a literature review of vesicovaginal leiomyomas, summarizing epidemiology, clinical presentation, diagnostic challenges, and management strategies, supported by tables and a comparative analysis.
This manuscript has clear clinical and educational value, as benign tumors in this location and age group can mimic malignancy, and the case highlights how multimodal imaging and biopsy can prevent overtreatment while preserving fertility.
Clinical description and methodology

    The history, exam, labs, and imaging are clearly presented. The diagnostic pathway (US → MRI → core biopsy) is appropriate and strengthens the case.
    Histology is well described and confirms leiomyoma. Still, adding h-caldesmon, ER/PR, or Ki-67 index would make the diagnosis more robust. Including a histological photomicrograph with a scale bar would be valuable.
    The surgical management (transvaginal enucleation) was well indicated and described. Please use generic terms for sutures rather than brand names.

Imaging

    MRI findings are described, but the statement that “restricted diffusion” is compatible with leiomyoma is problematic. Restricted diffusion often raises concern for malignancy. I strongly recommend reporting ADC values, detailing ROI placement and b-values, and ideally including an ADC map figure. This would clarify how the authors distinguished leiomyoma from sarcoma.

Literature review

    The review adds value but needs more transparency. Percentages such as “100% urinary symptoms” or “zero recurrences” are given, but it is not clear how many cases each number is based on. A PRISMA-style diagram and a supplementary case-by-case table (author, year, age, mass size, imaging, treatment, follow-up, outcome) are needed to support these data.
    Claims such as “the youngest reported case was 22 years” should be referenced to the exact source. Otherwise, please phrase cautiously (e.g., “to our knowledge, previous youngest reported…”).

References

    The list is broad and recent, but there are inconsistencies and duplicates (e.g., PubMed ID 30286580 appears twice; one entry “Gordts S.” is incomplete; some entries link to patient-information webpages). A full reference audit is needed: check formatting, remove duplicates, ensure all are peer-reviewed sources, and fix malformed DOIs.

Figures and tables

    Ultrasound and MRI figures need scale bars, adequate resolution, anonymization, and complete legends so they stand alone.
    Tables are informative but currently have formatting issues (overlap of text, abbreviations undefined). They must be cleaned and standardized.
    Figure 3 (comparative chart) should specify the denominators and data sources for each percentage shown.

Conclusions

    The conclusions are sound and supported by the case. They correctly emphasize multimodal diagnostics and fertility-preserving surgery. The limitations section should be expanded to highlight the single-case design, short follow-up of only 6 months, absence of molecular studies, and possible reporting bias in the literature.

Major comments

    Report quantitative ADC values and, if possible, add ADC map figures to clarify MRI findings.
    Improve the literature review transparency: PRISMA flow diagram + supplementary table of all cases.
    Audit and revise the reference list to remove duplicates and ensure accuracy and consistency.
    Improve the figures and tables: resolution, formatting, scale bars, legends, and independence from the text.
    Expand the limitations section as detailed above.
    Rewrite the Informed Consent and Funding statements clearly, removing template placeholders.

Minor comments

    Shorten overly long and repetitive sentences for better readability.
    Standardize terminology: β-hCG (not beta hCG), GnRH (not Gonadotropine), consistent hyphenation of “extrauterine.”
    Avoid absolute terms like “gold standard” unless supported by references. Simply state that cystoscopy/urethroscopy were performed to confirm integrity.

Comments and Suggestions for Authors
This is a clinically valuable case report of a very rare tumor in a young woman. The diagnostic process and fertility-preserving surgery are well described and relevant. To strengthen the manuscript, please:

    Add ADC values and maps to support the MRI interpretation.
    Make your literature review more transparent with a PRISMA diagram and supplementary case table.
    Audit your references for accuracy, consistency, and quality.
    Improve the formatting and quality of figures and tables.
    Expand the limitations.
    Simplify some sentences and perform minor English editing for clarity.

Comments on the Quality of English Language

The English is understandable, but at times wordy and repetitive. Minor professional editing would improve clarity, conciseness, and overall readability.

Author Response

Dear Reviewer,

We sincerely thank you for your thoughtful and constructive feedback on our manuscript. Your observations have been very valuable in helping us improve both the clarity and the scientific rigor of the paper. Please find attached below our responses and the corresponding revisions made. 

We are grateful for your insightful feedback, which has greatly improved the scientific quality and presentation of our work. We hope that the revised manuscript now addresses all concerns raised and provides a more transparent, precise, and clinically valuable contribution.

With kind regards,

On behalf of all authors .

Round 2

Reviewer 1 Report

Comments and Suggestions for Authors

The authors have addressed all of my questions

Author Response

Thank you very much for your positive feedback and for taking the time to review our work. We truly appreciate your thoughtful comments and guidance throughout the process, which helped us to substantially improve the manuscript.

Reviewer 2 Report

Comments and Suggestions for Authors

I appreciate the effort made to revise the manuscript and to expand it into a broader literature-based discussion. The topic remains clinically relevant and interesting, and the addition of imaging details (particularly the DWI/ADC description) provides value for readers in the field of gynecologic imaging. However, several aspects still require substantial improvement before the paper can meet the editorial and scientific standards of Diagnostics.

  1. Study design and methodology
    The manuscript is now described as a systematic review, but the methodology does not adhere to PRISMA requirements. There is no clear search strategy, inclusion/exclusion criteria, time frame, or bias assessment. If the authors cannot provide these essential elements, the paper should be redefined as a narrative review accompanying the case report. Presenting it as “systematic” without methodological compliance may be misleading.

  2. Introduction and background
    The introduction gives an overview of the condition but presents contradictory epidemiological figures (“fewer than 50 cases since 1933 / 1733 / 300 cases”). Please verify the historical data and unify the statement with a single, referenced number from a reliable source. Several references are secondary or web-based (e.g., Cleveland Clinic). Replace these with peer-reviewed primary studies or reviews.

  3. Case presentation and imaging findings
    The addition of diffusion-weighted imaging parameters (b values and ADC measurement) is valuable. However, it would strengthen the discussion to contextualize the ADC range with published data differentiating benign leiomyoma, atypical leiomyoma, and leiomyosarcoma, emphasizing the known overlap among these entities. Please also ensure that all MRI acquisition parameters are clearly and consistently described.

  4. Results and tables
    The comparative tables contain heterogeneous data extracted from disparate studies. Given the very small sample size, statistical analysis and p-values are not appropriate. These should be reformatted as descriptive summaries, indicating the number of reported cases and main features without inferential statistics.

  5. Ethical and editorial statements
    The Informed Consent Statement is incomplete and should be rewritten as:
    “Written informed consent was obtained from the patient for publication of this case report and the accompanying images.”
    Similarly, the Funding, Institutional Review Board, and Patents sections must be finalized and free of placeholders (“Please add…”). If no funding or patents apply, please explicitly state “Not applicable.”

  6. References
    The formatting must strictly follow Diagnostics reference style.

In summary, the manuscript addresses an uncommon and educational case with potential value for readers, but it still falls short in structure, methodological accuracy, and editorial quality. A thorough revision addressing the above points will considerably strengthen the scientific and formal integrity of the paper.

Comments on the Quality of English Language

The English language and overall editorial style require further improvement. The manuscript still contains multiple typographical artifacts (e.g., “Formatted:…”, duplicated words, double punctuation), inconsistent tense usage, and residual fragments from tracked changes. Some sentences are syntactically awkward or merged during revision, particularly in the “Results” and “Discussion” sections. A thorough professional editing by a native English scientific writer is strongly recommended to improve clarity, flow, and conformity with Diagnostics editorial standards.

Author Response

I hope this message finds you well.

Thank you very much for your thoughtful and constructive feedback on our manuscript. Based on your comments, we have revised the paper to address all the points raised. 

I am attaching the revised version of the manuscript for your kind review. I hope the changes meet your expectations and contribute to strengthening the quality of the work.

With gratitude for your time and support,

Sincerely,
On behalf of all co-authors

Round 3

Reviewer 2 Report

Comments and Suggestions for Authors

I would like to start by acknowledging the excellent work done in this revised version. The authors have responded rapidly and accurately to the comments raised in the previous review rounds, and the overall quality of the manuscript has improved substantially. The paper now reads clearly, follows the structure expected in Diagnostics, and provides an interesting and educational contribution on a very rare entity.

The introduction is well written and contextualizes the case appropriately, using updated and relevant references. The case description and imaging findings are now detailed and coherent, with adequate explanation of DWI and ADC parameters. 

The review of the literature is now much more robust and informative. I appreciate the inclusion of a schematic figure and the clear summary tables, which improve readability and provide practical value for clinicians.

All ethical sections have been corrected, and the manuscript now adheres to the journal’s structure. 

The English language is clear and professional, and the overall presentation is now of high quality.

In summary, this is a well-prepared, well-illustrated, and well-referenced case report with real educational value. I congratulate the authors for their prompt and thorough revision. 

Author Response

We sincerely thank the reviewer for their thoughtful and encouraging comments. We are very grateful for the time and effort dedicated to reviewing our manuscript and for recognizing the improvements made in this revised version. Your positive feedback regarding the clarity, structure, and educational value of our work is highly appreciated. We are pleased that the revisions have addressed your previous concerns and that the manuscript now meets the journal’s standards.

Thank you once again for your constructive input and kind words, which have greatly contributed to enhancing the quality of our paper.